ecology/evolution/palaeontology

Carboniferous, Bears Ears, heterodonty, canine, synapsid

**Author for correspondence:**
Adam K. Huttenlocker
e-mail: ahuttenlocker@gmail.com

# A Carboniferous synapsid with caniniform teeth and a reappraisal of mandibular size-shape heterodonty in the origin of mammals

Adam K. Huttenlocker[1,2], Suresh A. Singh[3],
Amy C. Henrici[2], Stuart S. Sumida[4]

[1]Department of Integrative Anatomical Sciences, University of Southern California, Los Angeles, CA 90033, USA
[2]Carnegie Museum of Natural History, Pittsburgh, PA 15213, USA
[3]School of Earth Sciences, University of Bristol, Bristol BS8 1RL, UK
[4]Department of Biology, California State University San Bernardino, San Bernardino, CA 92407, USA

AKH, 0000-0002-7335-3208

Heterodonty is a hallmark of early mammal evolution that originated among the non-mammalian therapsids by the Middle Permian. Nonetheless, the early evolution of heterodonty in basal synapsids is poorly understood, especially in the mandibular dentition. Here, we describe a new synapsid, *Shashajaia bermani* gen. et sp. nov., based on a well-preserved dentary and jaw fragments from the Carboniferous–Permian Halgaito Formation of southern Utah. *Shashajaia* shares with some sphenacodontids enlarged (canine-like) anterior dentary teeth, a dorsoventrally deep symphysis and low-crowned, subthecodont postcanines having festooned plicidentine. A phylogenetic analysis of 20 taxa and 154 characters places *Shashajaia* near the evolutionary divergence of Sphenacodontidae and Therapsida (Sphenacodontoidea). To investigate the ecomorphological context of Palaeozoic sphenacodontoid dentitions, we performed a principal component analysis based on two-dimensional geometric morphometrics of the mandibular dentition in 65 synapsids. Results emphasize the increasing terrestrialization of predator–prey interactions as a driver of synapsid heterodonty; enhanced raptorial biting (puncture/gripping) aided prey capture, but this behaviour was probably an evolutionary antecedent to more complex processing (shearing/tearing) of larger herbivore prey by the late Early to Middle Permian. The record of *Shashajaia* supports the notion that the predatory feeding ecology of sphenacodontoids emerged in palaeotropical western Pangea by late Carboniferous times.

# 1. Introduction

The fossil record of non-mammalian synapsids archives changes in tooth morphology that would eventually give rise to mammal-like heterodonty—size-shape differences across the toothrow, organized into distinct incisor, canine and postcanine dentitions [1,2] (figure 1). Importantly, the shapes and sizes of teeth allow inference of ancient predator–prey interactions, and investigations of size-shape variation along the toothrow in synapsids may therefore unveil patterns in the expansion of terrestrial vertebrate dietary niches. Notably, the Carboniferous–Permian (C–P) transition (ca 298.9 Ma) coincided with the proliferation of Earth's first herbivore-dominated communities, a trophic structure that is the basis for today's terrestrial ecosystems [4]. Changes observed in lower jaw structure from Carboniferous–Triassic times likewise reflected a diversification of food capture, manipulation and mastication processes among the earliest terrestrial herbivores and carnivores [5,6].

The definition of 'heterodonty' remains problematic. Classically, Simpson [1] discerned two arbitrary categories of heterodonty: (i) 'incipient heterodonty', which he limited to therapsids, noting foremost the size variations among the maxillary canines and their corresponding dentary teeth (figure 1b) and (ii) 'advanced heterodonty', signifying further differentiation in cusp patterns of the postcanine teeth in some premammalian cynodonts (premolars versus molars) [1,2]. Nevertheless, significant methodological challenges limit such categorical definitions. First, functional heterodonty has evolved numerous times in vertebrates, including within fishes [7,8] and various tetrapod groups [2,9–16]. As such, non-mammalian heterodont dentitions often do not form tooth families homologous to those of mammals, which has made quantitative morphometric approaches preferable to categorical approaches when describing size-shape variation among reptile dentitions, including in varanids and crocodylians [15,16]. Second, there have been relatively few attempts to quantify size-shape variation in simple, conical dentitions compared with the more complex multicusped dentitions of mammals (e.g. [17,18]) and some saurian groups (e.g. [19]).

Among synapsids, the earliest group to exhibit marked size-shape variation along the mandibular toothrow was probably the Sphenacodontoidea (figure 1b)—the common ancestor of the sail-backed sphenacodontids (e.g. *Dimetrodon*), therapsids and all their descendants. Whereas Permian sphenacodontids like *Dimetrodon* are widely regarded as the first large-bodied terrestrial apex carnivores [20–22], sphenacodontians were initially small-bodied faunivores during late Carboniferous and earliest Permian times (approx. 1–10 kg), including the Carboniferous *Haptodus* and *Ianthodon*, and the Permian *Palaeohatteria* and *Pantelosaurus* [3,23–28]. Spindler [29] added to these the Kasimovian-aged *Kenomagnathus scottae*, a possible congener of *Haptodus garnettensis* which is from the same locality in eastern Kansas. *Cutleria wilmarthi* from the undivided Cutler of western Colorado [30] was previously suggested to be a basal sphenacodontian akin to these forms [3,24,31] but has since been shown to be the basalmost sphenacodontid [28,32]. Recently, Brink *et al.* [21,33] demonstrated an underappreciated dental diversity among these early sphenacodontians, including variations in dental histology, serrations and cusp patterns that underlie important taxonomic differences and ecological diversity in the group. However, significant gaps in their fossil record and the rarity of Carboniferous sphenacodontians limit our interpretation of their ancestral dentition and its ecological context.

## 1.1. Present study

Here, we describe the sphenacodontian *Shashajaia bermani* gen. et sp. nov. from the C–P Halgaito Formation (Cutler Group) of southern Utah, USA and investigate its implications for the early evolution of mandibular heterodonty in synapsids. The material was collected from a conglomerate at the base of a multitaxic bonebed preserving faunal elements shared with Carboniferous assemblages in New Mexico [34,35], including *Sagenodus copeanus*, *Edaphosaurus*, *Ophiacodon navajovicus*, *Eryops*, *Sphenacodon* and limnoscelid diadectomorphs, among others. The vertebrate assemblages of the Cutler Group were some of the first terrestrial assemblages that included large-bodied vertebrate herbivores and specialist predators [36–38]. The dental morphology of the new taxon supports an expansion of tooth morphospace and pronounced size-shape heterodonty in the common ancestor of sphenacodontids and therapsids during late Carboniferous times. This ecomorphological diversification coincided with the late Palaeozoic remodelling of land-based food webs, new vertebrate dietary guilds, and fills a crucial gap in the synapsid fossil record.

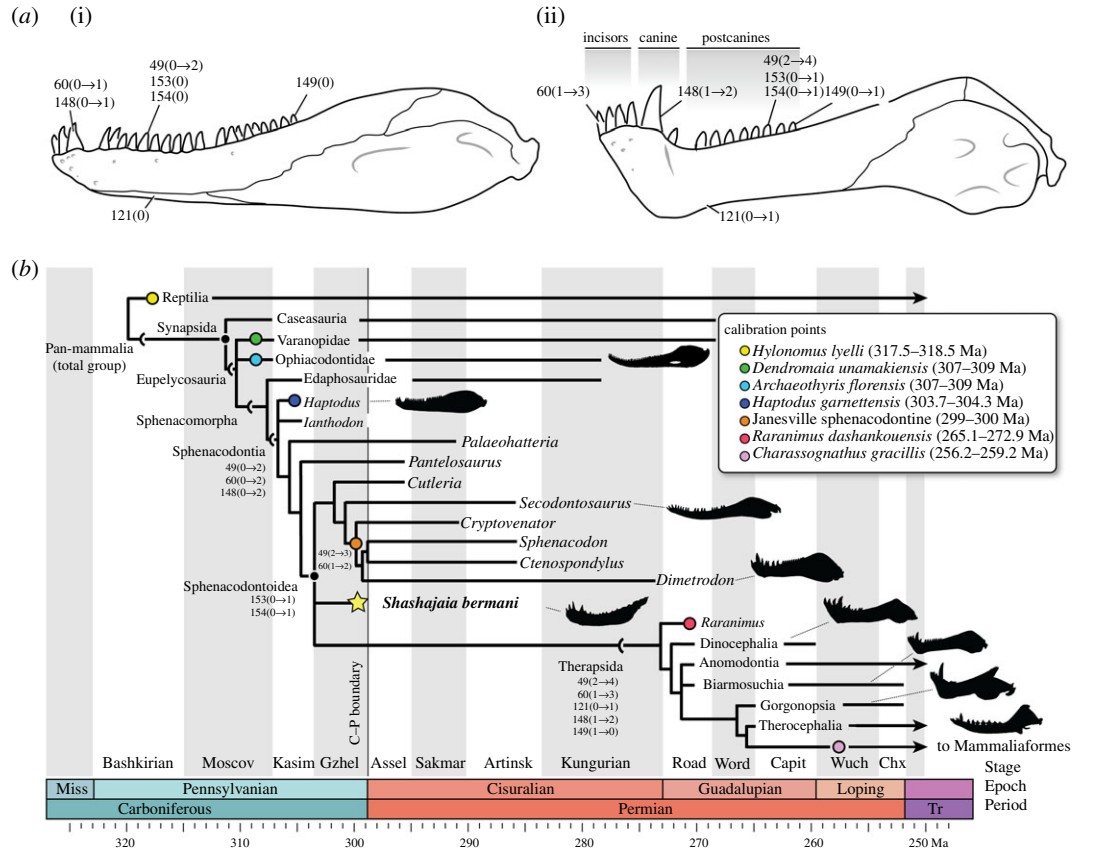

**Figure 1.** Evolution of mandibular and dental features in C–P synapsids. (*a*) Comparison of the mandible and dentition in a Carboniferous sphenacodontian (i) and a Middle Permian therapsid (ii). (i) Based on *Haptodus* (redrawn from [3]); (ii) based on *Biarmosuchus* (composite restoration of PIN 1758/2, 7, 8, and 307). (*b*) Time-calibrated phylogeny of the major clades of non-mammalian synapsids based on the parsimony analysis. Coloured internal and terminal nodes represent selected fossil calibration points that are tied to reliable geochronological ages. Star shows the position of the new taxon. Numbers 49–154 represent phylogenetically informative characters in the electronic supplementary material and their inferred DELTRAN state changes (in parentheses). Abbreviations: Artinsk, Artinskian; Assel, Asselian; Capit, Capitanian; Chx, Changxingian; Gzhel, Gzhelian; Kasim, Kasimovian; Loping, Lopingian; Miss, Mississippian; Moscov, Moscovian; Road, Roadian; Sakmar, Sakmarian; Tr, Triassic; Word, Wordian; Wuch, Wuchiapingian.

# 2. Material and methods

## 2.1. Fossil specimens

The fossils herein referred to *Shashajaia bermani* were collected from the basal conglomerate at the Birthday Bonebed locality in Valley of the Gods, SE Utah, Carnegie Museum (CM) locality no. 3345 [35]. The specimens consist of a well-preserved left dentary with dentition collected in 2019 and an unassociated dentary fragment collected from the same lens by the authors in 2015 [39,40]. The fossils were initially prepared mechanically using pneumatic tools and pin vice and were further inspected using computed tomography at the University of Southern California Molecular Imaging Center. The fossils were µCT-scanned on a GE Phoenix Nanotom at 22 µm (CM 96529) and 36 µm (CM 91209) resolution at 120 kV and 100 µA.

## 2.2. Ecomorphological analyses

### 2.2.1. Tooth size-shape heterodonty

Variations in mandibular tooth size and shape were assessed in *Shashajaia bermani* and 65 synapsids that lived from the Carboniferous–Triassic using the Type II landmark-based approach modified from D'Amore [15] and D'Amore *et al.* [16]. Major sampled groups included: Caseasauria (*N* = 7),

Varanopidae ($N = 4$), Ophiacodontidae ($N = 3$), Edaphosauridae ($N = 3$), non-therapsid sphenacodontians ($N = 11$, including *Shashajaia*); and the therapsid clades: Biarmosuchia ($N = 8$), Dinocephalia ($N = 6$), Anomodontia ($N = 4$), Gorgonopsia ($N = 4$), Therocephalia ($N = 11$) and Cynodontia ($N = 4$) (electronic supplementary material, tables S1 and S2). For each complete/undamaged mandibular tooth in each specimen, two-dimensional landmarks were digitized separately using high-resolution images of the mandible in lateral view with scale bar. In order to deal with the uncertainty of homologous landmarks, we applied two fixed landmarks at the base of each tooth and interpolated a curve comprising 28 semi-landmarks outlining each crown (electronic supplementary material, figure S5). Fixed landmarks were digitally applied in tpsDig2 [41] and semi-landmarks designated using tpsRelW [42]. A Procrustes alignment was then performed using tpsRelW [42] to standardize differences in image size and tooth orientation and to generate aligned coordinate data. The aligned landmark coordinate data were exported to R [43] for principal component analysis (PCA) using the geomorph package [44] to identify the principal axes of shape variation (see electronic supplementary material, Text). The resulting principal components (PCs) were used to examine patterns of variation along the toothrow and by clade. PC1 represents the majority of the variation, so its variance along the toothrow in each specimen was calculated and used as a measure of functional heterodonty.

### 2.2.2. Body size evolution

Because the canine dentition is interpreted to have facilitated carnivory in large predators like *Dimetrodon* and early therapsids [20], we tested whether therapsid-like heterodonty was driven in part by the expansion of large synapsid prey (especially herbivores) during the C–P transition. We estimated body size from a subset of 127 synapsids in which relatively complete linear measurements of skull and femur lengths were possible (electronic supplementary material, table S2). Body mass was estimated using a power function relationship between femur length and body mass (in kg) derived from a dataset of extant non-mammalian tetrapods published by Campione and Evans [45]. Resulting body size data were then compared with the tooth morphometric results in time series (figure 3*d*).

## 3. Description and discussion

### 3.1. Systematic palaeontology

Synapsida [46]
Eupelycosauria [47]
Sphenacodontia [20]
*Shashajaia bermani* gen. et sp. nov. (figure 2; electronic supplementary material, figures S1–S3).
*Etymology*—'Berman's Bear heart.' The genus name derives from the Navajo 'shash' (=bear) and 'ajai' (= heart). The species name honours David S Berman for his decades of research on fossils of sphenacodontians and others from the Bears Ears region of southern Utah, and which laid the foundation for the present study.

 *Holotype*—CM 96529 (Carnegie Museum of Natural History, Pittsburgh), an isolated left dentary preserving the dentition (figure 2; electronic supplementary material, figures S1 and S2).

 *Referred specimen*—CM 91209, partial dentary preserving portion of the postcanine toothrow (electronic supplementary material, figure S3).

 *Diagnosis*—Small non-mammalian sphenacodontian that can be distinguished from others by the unique combination of characters: slender, gently bowed dentary that deepens slightly anteriorly near symphysis; shallow, lateral groove positioned posterodorsally on dentary just below the postcanine toothrow; at least 24 lower tooth positions; anterior incisor- and canine-like dentary teeth consecutively increase in size posteriorly, with the fourth lower tooth separated by a short, concave diastema and positioned on a raised buttress; the remaining postcanine teeth are greatly reduced in height relative to the depth of the dentary (as in basal therapsids). Shares with *Haptodus*, *Ianthodon* and *Palaeohatteria* triangular, anteroposteriorly wide posterior cheek teeth that lack mesiodistal cutting edges. Shares with the sphenacodontids *Sphenacodon* and *Dimetrodon* festooned infolding of plicidentine.

### 3.2. Detailed description and discussion

The more complete of the two dentaries (CM 96529) is 12 cm long as preserved, with an estimated total jaw length of about 15 cm assuming similar proportions to *Palaeohatteria* and *Pantelosaurus* [28]. The main

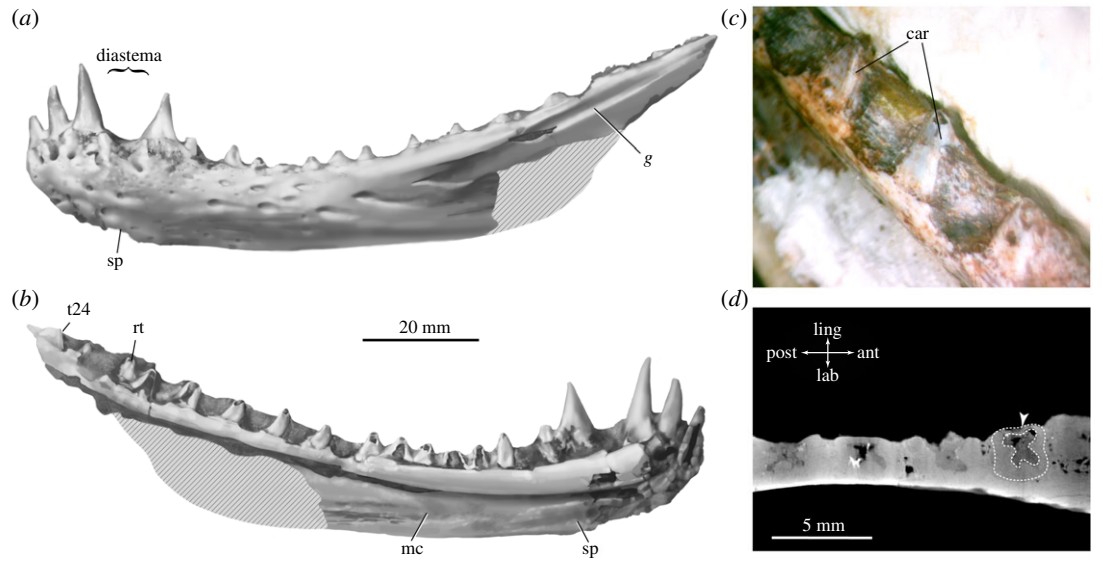

**Figure 2.** *Shashajaia bermani* gen. et sp. nov. Interpretive drawings of the holotypic mandible (CM 96529) in left lateral (*a*) and medial (*b*) views. (*c*) High-magnification photograph of postcanine teeth in oblique anteromedial view. (*d*) Horizontal tomographic slice taken at the level of the postcanines demonstrates festooned infolding of plicidentine (arrowhead). Abbreviations: car, carinae; g, groove; mc, fossa for Meckel's cartilage; rt, replacement tooth; sp, articular facet for splenial; t#, tooth number.

body is gently bowed and shallowest at mid-length so that the mid-level of the alveolar margin is substantially lower than either the incisiform teeth or the coronoid eminence. As in sphenacodontids and some therapsids, the dentary deepens dorsoventrally toward the symphysis and the anterior teeth are elevated on a thickened platform above the level of the postcanines [48]. The lateral surface of the dentary is textured with numerous large neurovascular pits that connect to an internal canal—the inferior alveolar nerve canal—which spans most of the length of the dentary to about the level of the caniniform tooth. The posterior portion of the ramus bears a shallow but distinctive lateral groove just ventrolateral to the alveolar margin in both specimens. The medial surface bears a smooth, elongated fossa below the alveolar shelf that would have contributed to the Meckelian canal, and an anteroventral facet that would have accepted the splenial (figure 2*b*; electronic supplementary material, figure S2). As in *Cutleria* and therapsids, but unlike sphenacodontines, the splenial exposure probably would have been limited near the symphysis (based on its facet on the medial surface of the dentary).

Among the more striking features of *Shashajaia* are the prominent heterodont dentition, with well-developed anterior canine-like teeth that are up to 2.5 times taller than the postcanines. There are at least 24 preserved tooth positions in total. All of the teeth appear to exhibit a subthecodont implantation, seated slightly deeper on the medial side than on the lateral side, and with festooned infolding of plicidentine at the attachment site forming a 'four-leaf clover' cross-sectional shape of the pulp cavity as in *Sphenacodon* and *Dimetrodon limbatus* [21] (figure 2*d*). Similar festooned plicidentine has been reported in indeterminate materials from the late Carboniferous Ada Formation of Oklahoma (Ghzelian stage) ([39]: fig. 11). The first three dentary teeth in *Shashajaia* consecutively increase in size so that the third is the tallest, reminiscent of the pattern in *Sphenacodon* in which the third tooth position usually accommodates the largest tooth (electronic supplementary material, figure S9) [49,50]. There is a short diastema (less than 1 cm) between the third and fourth tooth positions (caniniforms). A diastema has been observed previously between the third and fourth tooth positions in some specimens of *Haptodus garnettensis* ([3]: fig. 10) and an indeterminate sphenacodontian from the Sangre de Cristo Formation ([51]: fig. 4). The fourth (caniniform) tooth is situated on a raised buttress of alveolar bone and is at least twice the height of the succeeding postcanine teeth. As in sphenacodontids and therapsids, the postcanines are greatly reduced in height relative to the dorsoventral depth of the dentary (less than 40%). The more posterior cheek teeth lack the typical 'teardrop' shape of sphenacodontids and are instead more similar in outline to those of *Haptodus*, *Ianthodon* and *Palaeohatteria*. The teeth are triangular with a slightly crooked tip and anteroposteriorly wide bases that nearly come into contact with preserved neighbouring teeth (see [28]: fig. 4 and [27]: fig. 16). However, in *Shashajaia*, the crowns are comparatively lower in aspect ratio

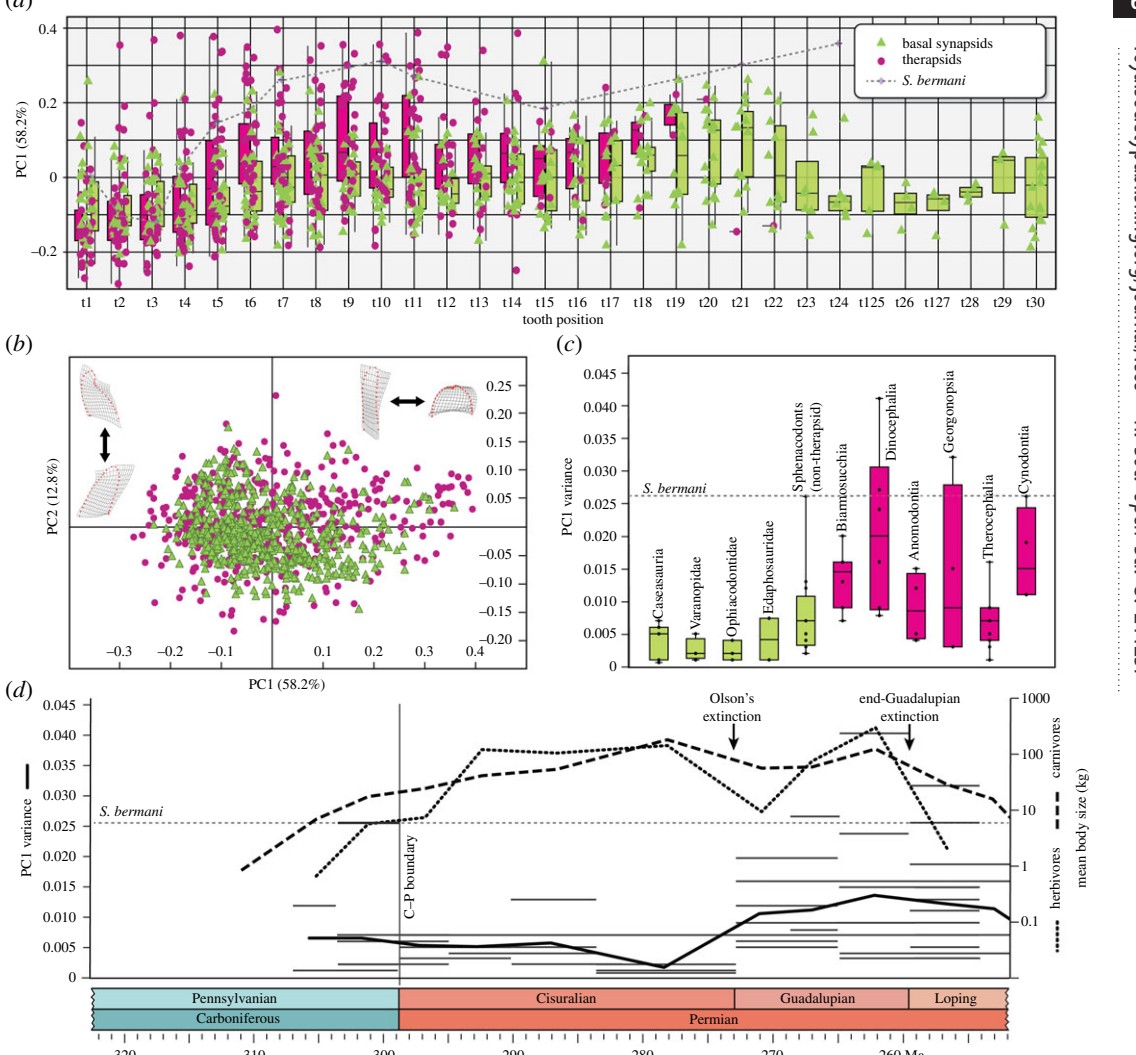

**Figure 3.** Results of ecomorphological analysis. (*a*) PC1 scores (obtained from PCA of tooth semi-landmarks) vary across the toothrow (t1-30) with therapsids (magenta) showing generally higher PC1 scores compared with the other basal synapsids (green), and more posterior teeth tending toward higher PC1 values. *Shashajaia* (grey dotted line) shows successively higher PC1 scores toward the back of the row, overlapping therapsid values. (*b*) Therapsid tooth shapes occupy a greater portion of the dental morphospace than those of the more basal synapsids, with a greater range of variation in PC1 along the *x*-axis—a proxy for tooth crown aspect ratio—and PC2 along the *y*-axis—a proxy for crown orientation and curvature. (*c*) Box-and-whiskers plot comparing PC1 variance in the major synapsid subclades, and demonstrating high PC1 variance in therapsids indicative of functional heterodonty along the toothrow. Grey dotted line represents the degree of heterodonty in *Shashajaia* which overlaps the highest therapsid values. (*d*) Time-series analysis of synapsid carnivore (dashed line) and herbivore (dotted line) mean body size and heterodonty (PC1 variance) through time (dicynodonts which lack a pre- and postcanine dentition are excluded). Horizontal bars indicate the PC1 variance of individual synapsid taxa whereas the solid line represents the mean.

and have a wider base, hyperbolic in outline. Distally, the teeth are medially compressed but are only weakly carinated and thus without mesiodistal cutting edges or serrations (figure 2*c*), suggesting they were probably not important for shearing or tearing.

### 3.2.1. Phylogenetic analysis

Comparisons to other sphenacodontians were further evaluated by parsimony analysis of 20 synapsid taxa and 154 morphological characters, updated from Fröbisch *et al.* [31] and Spindler *et al.* [28] and executed in PAUP 4.0a (build 167) [52]. The analysis recovered eight most parsimonious trees of 308 steps (consistency index, 0.6526; retention index, 0.8126) (see electronic supplementary material, Text). In the consensus tree, *Shashajaia bermani* resolves among the anatomically derived Sphenacodontoidea

in a polytomy with Sphenacodontidae and Therapsida (figure 1*b*). The taxon shares with all other sphenacodontians in the analysis: marginal teeth that are robust and sharp rather than peg-like (char. 49); and enlarged anterior dentary teeth more than 30% taller than the average tooth height (char. 60). Unlike the more basal sphenacodontians, *Shashajaia* shares with sphenacodontids and therapsids: splenial mostly exposed only medially (as in *Cutleria* and therapsids; char. 121); bases of the lower anterior (incisiform) teeth are on a raised bony platform elevated above the level of the more posterior teeth (char. 145); an enlarged caniniform tooth present on a separate buttress of alveolar bone (char. 148); and absolute crown heights of posterior dentary teeth are reduced to less than 40% the dorsoventral depth of the mandible (char. 153). Notably, the oldest geochronologically constrained sphenacodontid fossils are found in the approximately coeval late Carboniferous Janesville Formation (Admire Group) of the midcontinental USA [53] (figure 1*b*; electronic supplementary material, figure S4). Thus, *Shashajaia* fills an important morphologic and temporal gap between the basal haptodont-grade sphenacodontians and Sphenacodontoidea (figure 1*b*).

## 3.3. Morphometric results

The phylogenetic position of *Shashajaia* among Sphenacodontoidea presents an opportunity to assess the diversification of tooth morphospace in synapsids prior to the origin of therapsid-like heterodonty. Results of the morphometric analysis indicate that the first two PCs represent the overwhelming majority of tooth shape variation (71%) and were therefore used in subsequent plots to distinguish both (i) morphological variation along the toothrow (figure 3*a*; electronic supplementary material, figure S6) and (ii) overall variation within different taxonomic groups (figure 3*c*; electronic supplementary material, figures S7 and S8). PC1 encompassed the majority of tooth shape variation (58.2%) and chiefly reflects variation in tooth aspect ratio, which forms a continuum ranging from tall, slender (more negative) to short, stout (more positive) tooth crowns (figure 3*b*). This continuum along PC1 generally marks the sequential change in tooth shape from the front to the back of the toothrow (electronic supplementary material, figure S7) and shows a weak but negative correlation with tooth size in sphenacodontians (electronic supplementary material, figure S8). PC2 (12.8%) reflects overall tooth curvature, illustrating whether the curve manifests across the entire crown (more negative) or is concentrated towards the apex (more positive) (figure 3*b*). Overall, therapsids occupy a greater area of morphospace than more basal (pelycosaur-grade) synapsids (figure 3*a*,*b*). Dinocephalians exhibit the greatest morphospace occupation of all subclades, although therocephalians are more widely distributed across PC2 (electronic supplementary material, figure S7). Although sphenacodontians display the greatest dental disparity of all pelycosaur-grade synapsids, *Shashajaia*'s morphospace corresponds even more closely with therapsids as it shows a similarly broad distribution across PC1 (electronic supplementary material, figures S6 and S7).

In addition to filling a greater portion of the tooth morphospace, therapsids generally exhibited more positive PC1 values throughout the toothrow and with a significant increase (lower aspect ratio) toward the back of the row than in the more basal pelycosaur-grade synapsids where the teeth are comparatively more homodont (figure 3*a*; electronic supplementary material, figure S6). Variance in PC1 across an individual toothrow—a quantitative index for shape heterodonty—was therefore also greater in most therapsid clades than in basal synapsid groups (figure 3*c*); the highest therapsid PC1 variance was found among biarmosuchians, dinocephalians, gorgonopsians and cynodonts, and the lowest therapsid PC1 variance among non-dicynodont anomodonts and small therocephalian insectivores (e.g. *Tetracynodon*). Remarkably, PC1 variance in *Shashajaia* substantially overlapped the greatest therapsid values, probably reflecting the strong functional heterodonty of the tall, slender incisor and canine teeth versus the short, triangular postcanines. Size variation was loosely associated with variations in tooth aspect ratio, with the anterior teeth on average having a larger relative size than the more posterior teeth in both basal synapsids and therapsids. Primitively, the lower postcanine field of therapsids is known to show a marked decrease in tooth height relative to the overall jaw depth [48] beginning at about the level of the fifth or sixth tooth position (electronic supplementary material, figure S6), although sphenacodontids also exhibit comparatively short postcanine crowns relative to the dentary depth. Nevertheless, in the majority of basal synapsids, a functional tooth in any given position is not significantly smaller than its preceding tooth, but *Shashajaia* with its diminutive postcanines is more similar to the therapsids in this respect (electronic supplementary material, figures S6 and S7).

Time-series data (figure 3*d*) show a discordance between patterns of dental morphology and body size evolution in Palaeozoic synapsids. Mean body size gradually increased in both synapsid herbivores and carnivores from the late Carboniferous to Middle Permian with a short-term decrease

at the Early–Middle Permian boundary (figure 3d 'Olson's extinction'), corroborating similar findings by Brocklehurst & Brink [22] and Brocklehurst et al. [54]. Though initially lower on average than sympatric carnivores, mean herbivore size surpassed that of carnivores during the Early Permian, a possible ecological consequence of predator pressure on C–P herbivores and overall greater numbers of large herbivores during the late Early Permian. Nevertheless, expansion of body size disparity was not accompanied by similar increases in PC1 variance. In fact, mean PC1 variance was relatively stable or slightly decreased for each geologic stage from the Kasimovian to the Kungurian. Among late Carboniferous taxa, Shashajaia showed the highest PC1 variance (0.026). Among all of the taxa sampled, comparable PC1 variance was only re-encountered later in the Middle Permian therapsids (Capitanian stage), which included the largest known Palaeozoic synapsid herbivores and carnivores: the dinocephalians (100–1000 kg). Given this temporal lag and the inversion of large herbivores relative to carnivores, it is unlikely that regional specialization of tooth function in Shashajaia and the first therapsids were influenced by the same ecological factors. Moreover, the Halgaito assemblage from which Shashajaia originated includes abundant aquatic and semi-terrestrial taxa that proportionately outnumbered the fully terrestrial taxa (in contrast with the assemblage of the Permian Organ Rock Formation which was progressively more terrestrial) [35,55].

The comparisons outlined above bring into focus the ecological backdrop under which therapsid-like heterodonty evolved. During the C–P transition, herbivore richness and body size disparity dramatically increased [56] which, in addition to climate change, may have impacted plant species richness during Permian times [54]. Simultaneously, synapsid predators like the sphenacodontians showed increasing body size disparity [22,23], and functional innovations in their marginal dentition suggest underappreciated dietary diversity among the group ranging from generalist faunivores to more specialized apex predators [21]. While it is often assumed that sphenacodontids preyed on large synapsid herbivores, like Edaphosaurus [20], there is ample evidence that sphenacodontids originated as small-bodied faunivores—as in a documented predator–prey association between Dimetrodon milleri and the amphibian Zatrachys [20]. Shashajaia reflects this early sphenacodontian ecology, with its relatively gracile, upwardly curved dentary (figure 2) suggesting low stress, low power jaw functionality focused on speed and unsuited to extended struggles with prey during feeding [57]. The dental modification in Shashajaia probably compensates for the low-biting efficiency at the tip of the dentary, using instead the high velocity at this part of the jaw [6,7] alongside the enlarged anterior teeth to maximize the impact on the prey and penetrate deep into its flesh [40,58]. Given pelycosaur-grade synapsids were not particularly agile [20], this likely jaw function suggests Shashajaia was an ambush predator that fed on smaller animals, which could be quickly caught and swallowed without struggle. Thus, prior to the establishment of large herbivore-dominated tetrapod communities, as consumers shifted from aquatic to terrestrial feeding, C–P sphenacodontoids like Shashajaia most likely used their enlarged anterior dentition primarily for gripping and puncturing small to mid-sized prey, including perhaps small reptiles, amphibians and fishes [4,59,60]. Though the dentitions of Carboniferous taxa appear to have relatively smooth mesiodistal edges, the canine and antecanine dentition became modified at a later stage with serrations and denticles, forming a 'ziphodont' dentition. Ziphodonty in some large-bodied Dimetrodon species [21] and numerous therapsids [61] might indicate more efficient processing of protein in active terrestrial animals that had relatively greater metabolic requirements than the earliest sphenacodontians. Prior to this, the emphasis on prey capture may reflect increasing environmental heterogeneity as the Carboniferous drew to a close [62,63], with seasonal fluvial palaeoenvironments like that of the Birthday Bonebed [35] providing an ideal mix of terrestrial and semi-aquatic prey that allowed basal synapsid faunivores to experiment with varying their exposure to aquatic resources, enabling their trophic ecologies to become more firmly planted in the terrestrial realm.

Data accessibility. The electronic data are available in the electronic supplementary material, Dataset S1 and Dataset S2. Museum-curated specimens may be studied with permission from the host institution (CM). The data are provided in the electronic supplementary material [64].

Authors' contributions. A.K.H., A.C.H. and S.S.S. hold palaeontological permits for the study area and have conducted the fieldwork; A.K.H. prepared the specimens and composed the specimen illustrations; S.A.S. and A.K.H. designed the analytical approach and S.A.S. performed the morphometric analysis and produced figures; all authors wrote the paper.

Competing interests. The authors declare no competing interests.

Funding. A.K.H. was supported by a Discovery Pool grant from the Canyonlands Natural History Association.

Acknowledgements. We thank participants of the Natural History Museum of Utah Copper Club for finding the holotypic specimen in 2019. Interpretive drawings of CM 96529 were prepared by artist Stephanie Abramowicz, Natural History

Museum of Los Angeles County. We thank Tautis Skorka of the USC Molecular Imaging Center for scanning the specimens, and Megan Sims of the University of Kansas Vertebrate Paleontology Collections (KUVP) for additional comparative data. We also thank Neil Brocklehurst and one anonymous reviewer for their helpful suggestions for improving the manuscript. Fieldwork was conducted under BLM permit nos. UT12-005S and UT17-001S.

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
