## [Peer Review File · Royal Society Open Science]

Review History

RSOS-211237.R0 (Original submission)

Review form: Reviewer 1 (Neil Brocklehurst)

Is the manuscript scientifically sound in its present form?

Yes

Are the interpretations and conclusions justified by the results?

Yes

Is the language acceptable?

Yes

Do you have any ethical concerns with this paper?

No

Have you any concerns about statistical analyses in this paper?

Yes

Recommendation?

Accept with minor revision (please list in comments)

Comments to the Author(s)

Huttenlocker et al. describe a new species of Carboniferous synapsid with unusual dentition and present an analysis of the evolutionary history of heterodonty along the mammal stem, examining various hypotheses regarding its origin and consequences. I think this is an excellent paper and thoroughly recommend it for publication. The specimens themselves provide an interesting insight to synapsid diversity during the Carboniferous. The quantitative analyses of heterodonty are extremely innovative and have the potential to provide a methodological benchmark for broader analyses in future. My comments are pretty minor, and mostly deal with wording and suggestions for figures, but I do have a couple of analytical suggestions I will get the analytical issue out of the way first. It regards the time calibration of the tree, used for illustrative purposes in figure 1 and for the time series analysis (I think, see following paragraph). The methods used to time calibrate the tree is left a bit hazy; the authors mention seven calibration points used to constrain seven divergence dates, but give no details on how the other divergence dates are inferred/estimated. Maybe I'm missing something, due to the lack of detail regarding the time calibration method, but I find it a bit bizarre that the timescaling of the tree seems to have been based on these 7 selected calibration points for node ages, and question why the authors did not employ tip-dating approaches more usually employed for these datasets. The justification for the, provided in the supplementary text, is that there is uncertainty and circularity in the ages of many of the early amniote taxa, and they were following the best practices established by Parham et al. (2012), constraining divergence dates by independent, robustly dated anchors. I do not find these justifications convincing. The concern highlighted by the authors regarding ages of taxa may be accounted for in tip dating approaches (see below), and the cited paper is intended to establish best practices for node age calibrations for molecular analyses of extant taxa. For trees containing fossil taxa I really have doubts as to the appropriateness of this approach; it limits the data used to infer timescaling to a few taxa, one of which (*Raranimus*) included as a tip taxon in the phylogeny, yet is also, according to the table in the Supplementary text (no number given; table on page 13) used as the calibration from the origin of Therapsida. This is not appropriate methodology, and more robust approaches are available. Since the authors have a character taxon matrix, they could subject it to analysis using the Fossilised Birth Death model in MrBayes or Beast, which has become to a large extent standard practice for timescaling trees of fossil data. This approach allows the inclusion of node age priors, but uses the ages of all the tips (providing means to account for uncertainty in the ages of the tips, a concern highlighted by the authors) and the morphological data as well, to provide much more robust timescaling of the tree. If they authors would prefer to constrain the topology to that found in the parsimony analysis that can be done, or they could use a posteriori time scaling methods like Cal3 or the Hedman approach (Bapst 2013; Lloyd et al 2016).

A side note related to this point; in the supplementary text on page 13 the authors state that they use the timescaled tree in the time-series analysis. But no detail on how it was employed is provided either in the main text or the supplement. In fact, I was entirely unaware that the tree had been used in any of the analyses until I read the supplement! I was under the impression that the results of the time series analysis were based solely on observed taxa (and so was not planning on making a big deal of the time calibration, since I thought it was just for illustrative purposes). Where does the phylogeny come in? Were ancestral sizes/variances in PC scores inferred by ancestral state reconstructions? Or were ghost lineages included?

Once you have a robust time calibrated tree, one thing that could be good to either include as a supplementary figure or figure 1 would be to map your proxy for heterodonty in each taxon (PC1 variance) over the phylogeny, maybe using a heatmap of colours as provided by, for example, the

contMap function in phytools. This could show much more clearly the extent of the shift in heterodonty seen in therapsids, and to what extent Shashajaia has greater heterodonty than other contemporary synapsids.

A more general comment that I think needs to be considered throughout the manuscript is to be very clear what you are referring to when you use the word “heterodonty”. You make the distinction in the introduction between what Simpson described as incipient and advanced heterodonty, before, a couple of sentences later, using the term “functional heterodonty”, a term which seems pretty easy to understand (different teeth have different functions) but which you don’t make clear whether you’re limiting this to shape-based heterodonty or including purely size based heterodonty. After all, although more dramatic for therapsids, size-based heterodonty is likely primitive for amniotes as a whole, at least in the upper jaw (a caniniform tooth or region is present in limnoscelid and tseajaid diadectomorphs, Hylonomus and Brouffia on the reptile side, and the earliest eothyridids, ophiacodontids and sphenacomorphs on the synapsid side). And presumably these different sized teeth had different functions. Following from this, we come to an interesting point regarding your principal component analysis that I think needs to be made more clear: it is purely shape-based heterodonty; the Procrustes analysis removes any size. This is a bit of a rambling comment, so to strip it down to what I actually think needs to be done: make it clear in the intro whether “functional heterodonty” includes size, maybe point out that size based heterodonty is very widespread and likely primitive for amniotes; and clarify that your principal component analysis is restricted to shape-based heterodonty.

Some more minor points:

For the phylogenetic analysis, I think some more detail would be appreciated in the main text: how many MPTs, and what do the different MPTs show as alternative positions for Shashajaia? I know this can be gleaned from the supplement with the paup log, but it’s the sort of info that is useful and interesting and can be better explained in a couple of sentences.

There’s a couple of data points I would like some clarification on. First, Cryptovenator’s tip in figure 1 leapt out at me as it extended well into the early Permian (the table in the supplement gave its age as Asselian to Sakmarian). I was under the impression that the Remigiussberg formation was considered latest Gzhelian, maybe earliest asselian, with radiometric ages hovering around the 300MA mark (Königer et al. 2002; Fröbisch et al. 2011). Do you have a source for the later dates that you use?

Second, in figure 3d, there’s what looks like a taxon, with a PC1 variance of about 0.0075, that has an age apparently extending from the late Carboniferous to the end of the Permian. There is not, to the best of my knowledge, any taxon with a range, or range uncertainty, or even ghost lineage, covering that time. Is there an error in your ages somewhere, or is this multiple taxa with very similar PC1 variance values that just end up looking like a single taxon?

In the figures: I realise for short form papers space is at a premium, so I leave this up to the authors to decide whether or not this is a practical suggestion. But I don’t think figure 3a really adds much to the illustration of heterodonty and think instead Supplementary Figure S7 is by far the best illustration of different extents of heterodonty in different clades. The thing about figure 3a is that every point is a tooth, and there’s no way of knowing which teeth belong to the same taxa or even the same clade. So while we see that there is a greater variety of tooth-types within therapsids than in pelycosaurids, it doesn’t show us that therapsid species/clades are more heterodont. Figure S7 shows this much better (also showing that some clades, like edaphosaurids and varanopids) do show more heterodonty along PC2). Again, I know Fig S7 is a large, multipanel figure, and you can’t just replace figure 3a with it, and it might take up too much space, but if you’re able to bring it into the main text, I think it would be advisable.

Finally some wording and typos

Why is “heterodonty” in the first sentence of the abstract in quotation marks?

Section 2b, description of Ecomorphological analysis, page 7: “PC1 represents the greatest aspect of variation...” This is a tautology – PC1 always represents the greatest aspect of variation; it is defined as the axis of greatest variance. I assume you meant it represents the majority of the variation, since it accounts for more than 50% of the variability?

Throughout: the word diastem is used to describe the gap between the teeth. I don’t know if it was intentional to substitute diastem for diastema or not (I know diastem is a geological term), but diastema is what is conventionally used for gaps between teeth.

Neil Brocklehurst

Bapst, D. W. (2013). A stochastic rate-calibrated method for time-scaling phylogenies of fossil taxa. *Methods in Ecology and Evolution*, 4(8), 724-733.

Fröbisch, J., Schoch, R. R., Müller, J., Schindler, T., & Schweiss, D. (2011). A new basal sphenacodontid synapsid from the Late Carboniferous of the Saar-Nahe Basin, Germany. *Acta Palaeontologica Polonica*, 56(1), 113-120.

Königer, S., Lorenz, V., Stollhofen, H., & Armstrong, R. A. (2002). Origin, age and stratigraphic significance of distal fallout ash tuffs from the Carboniferous–Permian continental Saar–Nahe Basin (SW Germany). *International Journal of Earth Sciences*, 91(2), 341-356.

Lloyd, G. T., Bapst, D. W., Friedman, M., & Davis, K. E. (2016). Probabilistic divergence time estimation without branch lengths: dating the origins of dinosaurs, avian flight and crown birds. *Biology Letters*, 12(11), 20160609.

Review form: Reviewer 2

Is the manuscript scientifically sound in its present form?

Yes

Are the interpretations and conclusions justified by the results?

Yes

Is the language acceptable?

Yes

Do you have any ethical concerns with this paper?

No

Have you any concerns about statistical analyses in this paper?

No

Recommendation?

Accept with minor revision (please list in comments)

Comments to the Author(s)

This manuscript presents a new Carboniferous synapsid, *Shashajaia bermani* gen. et sp. nov., based on isolated jaw elements from the Halgaito Formation of southern Utah. The new taxon is described in detail and its relationships are analyzed, proposing its phylogenetic position in a polytomy at the base of Sphenacodontoidea. The taxon is characterized by the presence of

enlarged caniniform, anterior dentary teeth and as a result of the extremely short postcanine teeth also a pronounced size-related heterodonty. The authors take this as a starting point for quantifying heterodonty in the lower jaw using a 2D geometric morphometric approach based on 65 Paleozoic synapsids. According to the authors, the results suggest an increasing terrestrialization of predatory-prey interactions as the initial driver of synapsid heterodonty. In this case, however, I would disagree, as the authors state themselves that *Shashajaia* itself, as one of the earliest heterodont synapsids, likely would have preyed on small reptiles, amphibians, and fishes. In fact, the authors emphasize the independent evolution of this feature in *Shashajaia* and other sphenacodontoids. Maybe this could be slightly reworded or clarified in more detail. Other than that, the manuscript is very well written and presented and I have only very few and mainly editorial comments and corrections. Overall, I do not have any major objections to this study and its conclusions. I consider the manuscript essentially publishable as is and strongly recommend its publication in RSOS.

I only have some very minor, mainly editorial corrections and recommendations:

Page 4, Line 43 correct spelling of 'Pantelosaurus'

Page 4, Line 55 correct spelling of 'Fröbisch'

Page 8, Line 55 correct spelling of 'diastema' - also page 10, lines 36 and 38

Page 16, Line 15 why not use precanine instead of antecanine?

Decision letter (RSOS-211237.R0)

Dear Dr Huttenlocker

On behalf of the Editors, we are pleased to inform you that your Manuscript RSOS-211237 "A Carboniferous synapsid with caniniform teeth and a reappraisal of mandibular size-shape heterodonty in the origin of mammals" has been accepted for publication in Royal Society Open Science subject to minor revision in accordance with the referees' reports. Please find the referees' comments along with any feedback from the Editors below my signature.

Please submit your revised manuscript and required files (see below) no later than 7 days from today's (ie 14-Oct-2021) date. Note: the ScholarOne system will 'lock' if submission of the revision is attempted 7 or more days after the deadline. If you do not think you will be able to meet this deadline please contact the editorial office immediately.

on behalf of Professor Marcelo Sanchez (Associate Editor) and Kevin Padian (Subject Editor)
openscience@royalsociety.org

Associate Editor Comments to Author (Professor Marcelo Sanchez):

By clarifying the points raised by one of the reviewers and addressing the smaller edits in a revised version, this great contribution can be improved and hopefully published by our Journal.

Reviewer comments to Author:

Reviewer: 1

Comments to the Author(s)

Huttenlocker et al. describe a new species of Carboniferous synapsid with unusual dentition and present an analysis of the evolutionary history of heterodonty along the mammal stem, examining various hypotheses regarding its origin and consequences. I think this is an excellent paper and thoroughly recommend it for publication. The specimens themselves provide an interesting insight to synapsid diversity during the Carboniferous. The quantitative analyses of heterodonty are extremely innovative and have the potential to provide a methodological benchmark for broader analyses in future. My comments are pretty minor, and mostly deal with wording and suggestions for figures, but I do have a couple of analytical suggestions I will get the analytical issue out of the way first. It regards the time calibration of the tree, used for illustrative purposes in figure 1 and for the time series analysis (I think, see following paragraph). The methods used to time calibrate the tree is left a bit hazy; the authors mention seven calibration points used to constrain seven divergence dates, but give no details on how the other divergence dates are inferred/estimated. Maybe I'm missing something, due to the lack of detail regarding the time calibration method, but I find it a bit bizarre that the timescaling of the tree seems to have been based on these 7 selected calibration points for node ages, and question why the authors did not employ tip-dating approaches more usually employed for these datasets. The justification for the, provided in the supplementary text, is that there is uncertainty and circularity in the ages of many of the early amniote taxa, and they were following the best practices established by Parham et al. (2012), constraining divergence dates by independent, robustly dated anchors. I do not find these justifications convincing. The concern highlighted by the authors regarding ages of taxa may be accounted for in tip dating approaches (see below), and the cited paper is intended to establish best practices for node age calibrations for molecular analyses of extant taxa. For trees containing fossil taxa I really have doubts as to the appropriateness of this approach; it limits the data used to infer timescaling to a few taxa, one of which (*Raranimus*) included as a tip taxon in the phyogeny, yet is also, according to the table in the Supplementary text (no number given; table on page 13) used as the calibration from the origin of Therapsida. This is not appropriate methodology, and more robust approaches are available. Since the authors have a character taxon matrix, they could subject it to analysis using the Fossilised Birth Death model in MrBayes or Beast, which has become to a large extent standard practice for timescaling trees of fossil data. This approach allows the inclusion of node age priors, but uses the ages of all the tips (providing means to account for uncertainty in the ages of the tips, a concern highlighted by the authors) and the morphological data as well, to provide much more robust timescaling of the tree. If they authors would prefer to constrain the topology

to that found in the parsimony analysis that can be done, or they could use a postiori time scaling methods like Cal3 or the Hedman approach (Bapst 2013; Lloyd et al 2016).

A side note related to this point; in the supplementary text on page 13 the authors state that they use the timescaled tree in the time-series analysis. But no detail on how it was employed is provided either in the main text or the supplement. In fact, I was entirely unaware that the tree had been used in any of the analyses until I read the supplement! I was under the impression that the results of the time series analysis were based solely on observed taxa (and so was not planning on making a big deal of the time calibration, since I thought it was just for illustrative purposes). Where does the phylogeny come in? Were ancestral sizes/variances in PC scores inferred by ancestral state reconstructions? Or were ghost lineages included?

Once you have a robust time calibrated tree, one thing that could be good to either include as a supplementary figure or figure 1 would be to map your proxy for heterodonty in each taxon (PC1 variance) over the phylogeny, maybe using a heatmap of colours as provided by, for example, the contMap function in phytools. This could show much more clearly the extent of the shift in heterodonty seen in therapsids, and to what extent *Shashajaia* has greater heterodonty than other contemporary synapsids.

A more general comment that I think needs to be considered throughout the manuscript is to be very clear what you are referring to when you use the word "heterodonty". You make the distinction in the introduction between what Simpson described as incipient and advanced heterodonty, before, a couple of sentences later, using the term "functional heterodonty", a term which seems pretty easy to understand (different teeth have different functions) but which you don't make clear whether you're limiting this to shape-based heterodonty or including purely size based heterodonty. After all, although more dramatic for therapsids, size-based heterodonty is likely primitive for amniotes as a whole, at least in the upper jaw (a caniniform tooth or region is present in limnoscelid and tseajaid diadectomorphs, *Hylonomus* and *Brouffia* on the reptile side, and the earliest eothyridids, ophiacodontids and sphenacomorphs on the synapsid side). And presumably these different sized teeth had different functions. Following from this, we come to an interesting point regarding your principal component analysis that I think needs to be made more clear: it is purely shape-based heterodonty; the Procrustes analysis removes any size. This is a bit of a rambling comment, so to strip it down to what I actually think needs to be done: make it clear in the intro whether "functional heterodonty" includes size, maybe point out that size based heterodonty is very widespread and likely primitive for amniotes; and clarify that your principal component analysis is restricted to shape-based heterodonty.

Some more minor points:

For the phylogenetic analysis, I think some more detail would be appreciated in the main text: how many MPTs, and what do the different MPTs show as alternative positions for *Shashajaia*? I know this can be gleaned from the supplement with the paup log, but it's the sort of info that is useful and interesting and can be better explained in a couple of sentences.

There's a couple of data points I would like some clarification on. First, *Cryptovenator*'s tip in figure 1 leapt out at me as it extended well into the early Permian (the table in the supplement gave its age as Asselian to Sakmarian). I was under the impression that the *Remigiusberg* formation was considered latest Gzhelian, maybe earliest asselian, with radiometric ages hovering around the 300MA mark (Königer et al. 2002; Fröbisch et al. 2011). Do you have a source for the later dates that you use?

Second, in figure 3d, there's what looks like a taxon, with a PC1 variance of about 0.0075, that has an age apparently extending from the late Carboniferous to the end of the Permian. There is not, to the best of my knowledge, any taxon with a range, or range uncertainty, or even ghost lineage,

covering that time. Is there an error in your ages somewhere, or is this multiple taxa with very similar PC1 variance values that just end up looking like a single taxon?

In the figures: I realise for short form papers space is at a premium, so I leave this up to the authors to decide whether or not this is a practical suggestion. But I don't think figure 3a really adds much to the illustration of heterodonty and think instead Supplementary Figure S7 is by far the best illustration of different extents of heterodonty in different clades. The thing about figure 3a is that every point is a tooth, and there's no way of knowing which teeth belong to the same taxa or even the same clade. So while we see that there is a greater variety of tooth-types within therapsids than in pelycosaurs, it doesn't show us that therapsid species/clades are more heterodont. Figure S7 shows this much better (also showing that some clades, like edaphosaurids and varanopids)

do show more heterodonty along PC2). Again, I know Fig S7 is a large, multipanel figure, and you can't just replace figure 3a with it, and it might take up too much space, but if you're able to bring it into the main text, I think it would be advisable.

Finally some wording and typos

Why is "heterodonty" in the first sentence of the abstract in quotation marks?

Section 2b, description of Ecomorphological analysis, page 7: "PC1 represents the greatest aspect of variation..." This is a tautology - PC1 always represents the greatest aspect of variation; it is defined as the axis of greatest variance. I assume you meant it represents the majority of the variation, since it accounts for more than 50% of the variability?

Throughout: the word diastem is used to describe the gap between the teeth. I don't know if it was intentional to substitute diastem for diastema or not (I know diastem is a geological term), but diastema is what is conventionally used for gaps between teeth.

Neil Brocklehurst

Bapst, D. W. (2013). A stochastic rate-calibrated method for time-scaling phylogenies of fossil taxa. *Methods in Ecology and Evolution*, 4(8), 724-733.

Fröbisch, J., Schoch, R. R., Müller, J., Schindler, T., & Schweiss, D. (2011). A new basal sphenacodontid synapsid from the Late Carboniferous of the Saar-Nahe Basin, Germany. *Acta Palaeontologica Polonica*, 56(1), 113-120.

Königer, S., Lorenz, V., Stollhofen, H., & Armstrong, R. A. (2002). Origin, age and stratigraphic significance of distal fallout ash tuffs from the Carboniferous-Permian continental Saar-Nahe Basin (SW Germany). *International Journal of Earth Sciences*, 91(2), 341-356.

Lloyd, G. T., Bapst, D. W., Friedman, M., & Davis, K. E. (2016). Probabilistic divergence time estimation without branch lengths: dating the origins of dinosaurs, avian flight and crown birds. *Biology Letters*, 12(11), 20160609.

Reviewer: 2

Comments to the Author(s)

This manuscript presents a new Carboniferous synapsid, *Shashajaia bermani* gen. et sp. nov., based on isolated jaw elements from the Halgaito Formation of southern Utah. The new taxon is described in detail and its relationships are analyzed, proposing its phylogenetic position in a polytomy at the base of Sphenacodontidae. The taxon is characterized by the presence of enlarged caniniform, anterior dentary teeth and as a result of the extremely short postcanine teeth also a pronounced size-related heterodonty. The authors take this as a starting point for quantifying heterodonty in the lower jaw using a 2D geometric morphometric approach based on 65 Paleozoic synapsids. According to the authors, the results suggest an increasing

terrestrialization of predatory-prey interactions as the initial driver of synapsid heterodonty. In this case, however, I would disagree, as the authors state themselves that Shashajaia itself, as one of the earliest heterodont synapsids, likely would have preyed on small reptiles, amphibians, and fishes. In fact, the authors emphasize the independent evolution of this feature in Shashajaia and other sphenacodontoids. Maybe this could be slightly reworded or clarified in more detail. Other than that, the manuscript is very well written and presented and I have only very few and mainly editorial comments and corrections. Overall, I do not have any major objections to this study and its conclusions. I consider the manuscript essentially publishable as is and strongly recommend its publication in RSOS.

I only have some very minor, mainly editorial corrections and recommendations:

Page 4, Line 43 correct spelling of 'Pantelosaurus'

Page 4, Line 55 correct spelling of 'Fröbisch'

Page 8, Line 55 correct spelling of 'diastema' – also page 10, lines 36 and 38

Page 16, Line 15 why not use precanine instead of antecanine?

===PREPARING YOUR MANUSCRIPT===

===PREPARING YOUR REVISION IN SCHOLARONE===

Author's Response to Decision Letter for (RSOS-211237.R0)

See Appendix A.

Decision letter (RSOS-211237.R1)

Dear Dr Huttenlocker,

I am pleased to inform you that your manuscript entitled "A Carboniferous synapsid with caniniform teeth and a reappraisal of mandibular size-shape heterodonty in the origin of mammals" is now accepted for publication in Royal Society Open Science.

Kind regards,
Royal Society Open Science Editorial Office
Royal Society Open Science

on behalf of Professor Marcelo Sanchez (Associate Editor) and Kevin Padian (Subject Editor)
openscience@royalsociety.org

Appendix A

Response to Referees *

Huttenlocker et al. "A Carboniferous synapsid with caniniform teeth and a reappraisal of mandibular size-shape heterodonty in the origin of mammals"

* Author responses are included below in blue text.

Reviewer: 1

Comments to the Author(s)

The methods used to time calibrate the tree is left a bit hazy; the authors mention seven calibration points used to constrain seven divergence dates, but give no details on how the other divergence dates are inferred/estimated. Maybe I'm missing something, due to the lack of detail regarding the time calibration method, but I find it a bit bizarre that the timescaling of the tree seems to have been based on these 7 selected calibration points for node ages, and question why the authors did not employ tip-dating approaches more usually employed for these datasets. The justification for the, provided in the supplementary text, is that there is uncertainty and circularity in the ages of many of the early amniote taxa, and they were following the best practices established by Parham et al. (2012), constraining divergence dates by independent, robustly dated anchors. I do not find these justifications convincing. The concern highlighted by the authors regarding ages of taxa may be accounted for in tip dating approaches ...

Unchanged (clarification provided here): The reviewer raises two critiques here: (1) Is the "best practices" method of Parham et al. applicable or useful to morphological trees and (2) Why not use the time-scaled tree for more sophisticated analysis of character rates and modes? First, taking our discussion in context, the main issue highlighted by *Shashajaia* is the minimum divergence time of Sphenacodontidae + Therapsida and the inferred ancestral state of its dentition. As we argue in the ESM, while the fragmentary material of sphenacodontians from Colorado and Kansas cannot be used as "tip taxa," an apomorphy-based approach can nonetheless be used to better constrain divergence times of internal nodes when higher taxa are diagnosable and are associated with independent geochronological controls (as in the Janesville sphenacodontid from Kansas, which is associated with marine biostratigraphic data from the same beds). These "best practices" were recommended by Parham et al. for independently constraining molecular trees, but are equally as applicable to assigning reliable and independent divergence times for morphological trees. Lastly, we appreciate the reviewer's suggestion for additional analyses that might shed further light on the rates and mode of character evolution in the synapsid tree, and we plan to investigate further in a follow-up paper for a specialist journal.

A side note related to this point; in the supplementary text on page 13 the authors state that they use the timescaled tree in the time-series analysis. But no detail on how it was employed is provided either in the main text or the supplement ... Where does the phylogeny come in?

Were ancestral sizes/variances in PC scores inferred by ancestral state reconstructions? Or were ghost lineages included?

Unchanged (clarification provided here): The phylogeny is shown in Figure 1b, and temporal calibrations were utilized to assess the age of *Shashajaia* in relation to other synapsids, and to demonstrate increasing mandibular heterodonty by latest Carboniferous times. Numbers 49–154 in Fig. 1b represent phylogenetically informative characters in the ESM Supplementary Information and their inferred DELTRAN state changes.

A more general comment that I think needs to be considered throughout the manuscript is to be very clear what you are referring to when you use the word “heterodonty”. You make the distinction in the introduction between what Simpson described as incipient and advanced heterodonty, before, a couple of sentences later, using the term “functional heterodonty”, a term which seems pretty easy to understand (different teeth have different functions) but which you don’t make clear whether you’re limiting this to shape-based heterodonty or including purely size based heterodonty.

Change made. We emphasize that heterodonty may encompass “size and shape” differences. Although the scaling effect of the Procrustes analysis corrects for any influence of size on the dataset, tooth aspect ratio (PC1) is generally higher in proportionately higher-crowned teeth, so these metrics are not necessarily independent as the reviewer suggests. We illustrate these separately in the online ESM and Figures S7 and new Figure S8 in which the more negative PC1 data points (higher aspect ratio) frequently exhibit a larger relative tooth size.

Some more minor points:

For the phylogenetic analysis, I think some more detail would be appreciated in the main text: how many MPTs, and what do the different MPTs show as alternative positions for *Shashajaia*? I know this can be gleaned from the supplement with the paup log, but it’s the sort of info that is useful and interesting and can be better explained in a couple of sentences.

Change made. Details added on p. 11 and reference to ESM added.

There’s a couple of data points I would like some clarification on. First, *Cryptovenator*’s tip in figure 1 leapt out at me as it extended well into the early Permian (the table in the supplement gave its age as Asselian to Sakmarian). I was under the impression that the Remigiusberg formation was considered latest Gzhelian, maybe earliest asselian, with radiometric ages hovering around the 300MA mark (Königer et al. 2002; Fröbisch et al. 2011). Do you have a source for the later dates that you use?

Unchanged (clarification provided here): SHRIMP dates are generally not as precise as TIMS dates, but the Königer study which the reviewer cites above places the maximum depositional age of the unit at approximately 297 Ma or younger. Given that the GSSP of the C-P boundary is 298.9 Ma, this suggests that *Cryptovenator* is roughly Asselian in age (early Permian), which

matches its position in our Figure 1. The Janesville sphenacodontine is geologically older and thus represents the oldest datum for Sphenacodontinae shown in Fig. 1 and the ESM Supplementary Information Text: Temporal Calibration Points and Justification.

Second, in figure 3d, there's what looks like a taxon, with a PC1 variance of about 0.0075, that has an age apparently extending from the late Carboniferous to the end of the Permian. There is not, to the best of my knowledge, any taxon with a range, or range uncertainty, or even ghost lineage, covering that time. Is there an error in your ages somewhere, or is this multiple taxa with very similar PC1 variance values that just end up looking like a single taxon?

Unchanged (clarification provided here): Yes, these are multiple taxa in different stages with similar PC1 variance.

In the figures: I realise for short form papers space is at a premium, so I leave this up to the authors to decide whether or not this is a practical suggestion. But I don't think figure 3a really adds much to the illustration of heterodonty and think instead Supplementary Figure S7 is by far the best illustration of different extents of heterodonty in different clades. The thing about figure 3a is that every point is a tooth, and there's no way of knowing which teeth belong to the same taxa or even the same clade. So while we see that there is a greater variety of tooth-types within therapsids than in pelycosaurids, it doesn't show us that therapsid species/clades are more heterodont. Figure S7 shows this much better (also showing that some clades, like edaphosaurids and varanopids) do show more heterodonty along PC2). Again, I know Fig S7 is a large, multipanel figure, and you can't just replace figure 3a with it, and it might take up too much space, but if you're able to bring it into the main text, I think it would be advisable.

Unchanged (clarification provided here): We are currently expanding these analyses for a larger quantitative study in a specialist journal. In the meantime, information requested by the reviewer is presented in Fig. S7, as this provides "specialist" information that clarifies finer-scale patterns within the different synapsid subgroups. Given the Royal Society's broad readership and our target audience for the *Shashajia* paper, we prefer to keep these taxonomic details in the supplement for now.

Finally some wording and typos

Why is "heterodonty" in the first sentence of the abstract in quotation marks?

Change made. Quotes removed.

Section 2b, description of Ecomorphological analysis, page 7: "PC1 represents the greatest aspect of variation..." This is a tautology – PC1 always represents the greatest aspect of variation; it is defined as the axis of greatest variance. I assume you meant it represents the majority of the variation, since it accounts for more than 50% of the variability?

Change made. Clarified to state "majority of the variation."

Throughout: the word diastem is used to describe the gap between the teeth. I don't know if it was intentional to substitute diastem for diastema or not (I know diastem is a geological term), but diastema is what is conventionally used for gaps between teeth.

Change made. "Diastem" changed to "diastema" and checked throughout.

Neil Brocklehurst

Reviewer: 2

Comments to the Author(s)

This manuscript presents a new Carboniferous synapsid, *Shashajaia bermani* gen. et sp. nov., based on isolated jaw elements from the Halgaito Formation of southern Utah. The new taxon is described in detail and its relationships are analyzed, proposing its phylogenetic position in a polytomy at the base of Spheancodontoidea. The taxon is characterized by the presence of enlarged caniniform, anterior dentary teeth and as a result of the extremely short postcanine teeth also a pronounced size-related heterodonty. The authors take this as a starting point for quantifying heterodonty in the lower jaw using a 2D geometric morphometric approach based on 65 Paleozoic synapsids. According to the authors, the results suggest an increasing terrestrialization of predatory-prey interactions as the initial driver of synapsid heterodonty. In this case, however, I would disagree, as the authors state themselves that *Shashajaia* itself, as one of the earliest heterodont synapsids, likely would have preyed on small reptiles, amphibians, and fishes. In fact, the authors emphasize the independent evolution of this feature in *Shashajaia* and other sphenacodontoids. Maybe this could be slightly reworded or clarified in more detail.

Change made. Wording changed and clarified on page 2 where this criticism mainly applies. We agree with the reviewer and iterate that this point is made explicit on pages 14-15 in which we state *"it is unlikely that regional specialization of tooth function in Shashajaia and the first therapsids were influenced by the same ecological factors. Moreover, the Halgaito assemblage from which Shashajaia originated includes abundant aquatic and semi-terrestrial taxa that proportionately outnumbered the fully terrestrial taxa (in contrast to the assemblage of the Permian Organ Rock Formation which was progressively more terrestrial)."*

Other than that, the manuscript is very well written and presented and I have only very few and mainly editorial comments and corrections. Overall, I do not have any major objections to this study and its conclusions. I consider the manuscript essentially publishable as is and strongly recommend its publication in RSOS.

We thank the reviewer for their encouragement.

I only have some very minor, mainly editorial corrections and recommendations:

Page 4, Line 43 correct spelling of 'Pantelosaurus'

Change made.

Page 4, Line 55 correct spelling of 'Fröbisch'

Change made.

Page 8, Line 55 correct spelling of 'diastema' – also page 10, lines 36 and 38

Change made.

Page 16, Line 15 why not use precanine instead of antecanine?

Clarification (unchanged): Antecanine is the more appropriate term in this context. "Antecanine" means "before" the canine, encompassing all of the teeth in front of the canines, including the incisors and precanines. In synapsid dental terminology, "precanine" only includes the precanine maxillary teeth in the upper dentition, which are not preserved here.